# Dual-localized PPTC7 limits mitophagy through proximal and dynamic interactions with BNIP3 and NIX

Lianjie Wei[1], Mehmet Oguz Gok[2], Jordyn D Svoboda[1], Keri-Lyn Kozul[1], Merima Forny[1], Jonathan R Friedman[2], Natalie M Niemi[1]

**PPTC7 is a mitochondrial-localized phosphatase that suppresses BNIP3- and NIX-mediated mitophagy, but the mechanisms underlying this regulation remain ill-defined. Here, we demonstrate that loss of PPTC7 upregulates BNIP3 and NIX post-transcriptionally and independent of HIF-1α stabilization. Loss of PPTC7 prolongs the half-life of BNIP3 and NIX while blunting their accumulation in response to proteasomal inhibition, suggesting that PPTC7 promotes the ubiquitin-mediated turnover of BNIP3 and NIX. Consistently, overexpression of PPTC7 limits the accumulation of BNIP3 and NIX protein levels, which requires an intact catalytic motif but is surprisingly independent of its targeting to mitochondria. Consistently, we find that PPTC7 is dual-localized to the outer mitochondrial membrane and the matrix. Importantly, anchoring PPTC7 to the outer mitochondrial membrane is sufficient to blunt BNIP3 and NIX accumulation, and proximity labeling and fluorescence co-localization experiments demonstrate that PPTC7 dynamically associates with BNIP3 and NIX within the native cellular environment. Collectively, these data reveal that a fraction of PPTC7 localizes to the outer mitochondrial membrane to promote the proteasomal turnover of BNIP3 and NIX, limiting basal mitophagy.**

## Introduction

Mitophagy, or mitochondrial-specific autophagy, is a conserved organellar quality control process that promotes the selective turnover of damaged or superfluous mitochondria (Pickles et al, 2018; Uoselis et al, 2023). During mitophagy, mitochondria are selectively tagged for degradation by the activation of either ubiquitin-associated pathways or specific mitophagy receptors. Multiple human diseases result from mutations within mitophagy-associated genes, such as Parkinson's disease (Kitada et al, 1998; Valente et al, 2004; Zimprich et al, 2011) and amyotrophic lateral sclerosis (Maruyama et al, 2010). Most of these disease-associated mutations decrease the efficiency of mitophagy, suggesting that enhancing mitophagy may promote mitochondrial function and alleviate various human pathologies (Lee & Kim, 2014; Mishra & Thakur, 2023; Wang et al, 2023). Interestingly, however, recent studies show that unrestrained mitophagy may also trigger pathophysiology, particularly by disrupting the regulation of the mitophagy receptors BNIP3 and NIX (Bonnen et al, 2013; Gai et al, 2013; Cao et al, 2023; Elcocks et al, 2023; Nguyen-Dien et al, 2023).

BNIP3 and NIX have long been associated with mitochondrial turnover. Well-characterized as transcriptional targets of hypoxia inducible factor 1α (HIF-1α), BNIP3 and NIX are upregulated during hypoxia to decrease mitochondrial content due to limited oxygen availability (Ney, 2015). NIX promotes mitochondrial clearance during erythropoiesis to prevent mature red blood cells from consuming the oxygen they carry before it is delivered to distal tissues (Schweers et al, 2007). Similarly, BNIP3 and NIX induce mitochondrial turnover and subsequent metabolic reprogramming in various models of cellular differentiation, including neurons (Ordureau et al, 2021), myoblasts (Sin et al, 2016), and cardiomyocytes (Esteban-Martínez & Boya, 2018; Zhao et al, 2020). These studies demonstrate that BNIP3 and NIX can potently decrease mitochondrial content across cell types, indicating the levels of these mitophagy receptors must be tightly regulated to prevent excessive mitochondrial clearance. Consistently, recent studies have shown that loss of the mitochondrial E3 ubiquitin ligase FBXL4 unleashes BNIP3- and NIX-mediated mitophagy, leading to decreased mitochondrial protein levels, mtDNA depletion, and perinatal lethality in mice (Alsina et al, 2020; Cao et al, 2023; Chen et al, 2023; Elcocks et al, 2023; Nguyen-Dien et al, 2023). Mutations in human *FBXL4* cause mitochondrial DNA depletion syndrome 13 (MTDPS13) a severe pathology characterized by encephalopathy, stunted growth, and metabolic deficiencies (Bonnen et al, 2013; Gai et al, 2013; Dai et al, 2017; Ballout et al, 2019). Importantly, these human mutations disrupt the ability of FBXL4 to promote BNIP3 and NIX turnover (Cao et al, 2023; Elcocks et al, 2023; Nguyen-Dien et al, 2023), suggesting that excessive BNIP3 and NIX accumulation constitutes a substantial organismal liability. Despite these compelling data, the mechanisms restraining these mitophagy receptors under basal conditions have not been fully defined.

[1]Department of Biochemistry & Molecular Biophysics, Washington University School of Medicine in St. Louis, St. Louis, MO, USA  [2]Department of Cell Biology, University of Texas Southwestern Medical Center, Dallas, TX, USA

Correspondence: niemi@wustl.edu

We previously identified the mitochondrial-resident protein phosphatase PPTC7 as a regulator of BNIP3- and NIX-mediated mitophagy (Niemi et al, 2023). KO of *Pptc7* in mice led to fully penetrant perinatal lethality concomitant to metabolic defects, including hypoketotic hypoglycemia (Niemi et al, 2019). Strikingly, tissues and cells isolated from *Pptc7* KO animals showed robustly decreased mitochondrial protein levels as well as consistently elevated BNIP3 and NIX protein expression (Niemi et al, 2019), indicating that unchecked BNIP3 and NIX expression may drive mitochondrial loss through excessive mitophagy. Indeed, KO of *Bnip3* and *Nix* within the *Pptc7* KO background largely rescues mitochondrial protein expression when suppressing mitophagy (Niemi et al, 2023). In addition, we found that BNIP3 and NIX are hyperphosphorylated in *Pptc7* KO systems and that PPTC7 can directly interact with BNIP3 and NIX to facilitate their dephosphorylation in vitro (Niemi et al, 2023). These data demonstrate that PPTC7 acts as a critical negative regulator of BNIP3- and NIX-mediated mitophagy. However, the precise molecular mechanism(s) by which PPTC7 influences BNIP3 and NIX protein levels and mitophagic flux remain unclear, particularly given that these proteins reside in separate mitochondrial compartments (Rhee et al, 2013; Hung et al, 2017). Here, we use a combination of biochemical and cellular assays to demonstrate that PPTC7 proximally and dynamically interacts with BNIP3 and NIX to promote their turnover and limit basal receptor-mediated mitophagy.

## Results

### PPTC7 regulates BNIP3 and NIX post-transcriptionally and independent of HIF-1α

We previously reported that BNIP3 and NIX were significantly upregulated in tissues and cells derived from *Pptc7*$^{-/-}$ mice (Niemi et al, 2019, 2023). Consistently, *Pptc7* KO MEFs (Fig 1A) and *PPTC7* KO 293T cells (Fig 1B) showed elevated expression of BNIP3 and NIX relative to wild-type cells. To understand the mechanisms underlying BNIP3 and NIX upregulation upon *PPTC7* loss, we investigated the involvement of HIF-1α in mediating this response. *BNIP3* and *BNIP3L* (gene name of NIX) are well-established transcriptional targets of HIF-1α in conditions of hypoxia, as well as pseudohypoxia through various pharmacological stimuli (e.g., the iron chelators deferoxamine [DFO] and deferiprone [DFP] as well as cobalt chloride [Wang & Semenza, 1993a, 1993b; Allen et al, 2013]). We thus hypothesized that the increase in BNIP3 and NIX protein levels in *PPTC7* KO cells may be due to elevated HIF-1α activity. To test this, we immunoblotted for HIF-1α expression in wild-type and *PPTC7* KO cells in both basal conditions as well as in the presence of bafilomycin A1 (Baf-A1), a compound previously shown to stabilize HIF-1α (Hubbi et al, 2013). These experiments revealed no differences in HIF-1α protein expression between wild-type and *PPTC7* KO cells across tested conditions (Fig 1C). In addition, we found no significant changes in the abundance of select HIF-regulated proteins relative to other proteins across our previously collected proteomics datasets in *Pptc7* KO mouse

tissues (Niemi et al, 2019) (Fig S1A). We next tested whether BNIP3 and NIX were transcriptionally upregulated but found no significant differences in *BNIP3* or *BNIP3L* mRNA levels between wild-type and *PPTC7* KO 293T cells (Fig 1D). Consistently, we found that the transient transfection of plasmids encoding myc-BNIP3 or FLAG-NIX led to increased protein expression in *PPTC7* KO 293T cells relative to wild-type 293T cells (Fig S1B and C). As these plasmids are driven by the same CMV promoter in both wild-type and *PPTC7* KO cells, the elevated BNIP3 and NIX protein expression seen in *PPTC7* KO likely occurs independent of transcriptional changes. Together, these data indicate that PPTC7 influences BNIP3 and NIX protein expression post-transcriptionally and is independent of HIF-1α.

The HIF-1α-independent upregulation of BNIP3 and NIX in *PPTC7* KO cells suggests that HIF-1α and PPTC7 regulate BNIP3 and NIX through parallel pathways. If true, we hypothesized that loss of *PPTC7* would not alter *BNIP3* and *BNIP3L* transcriptional induction upon HIF-1α activation, but would promote an additive increase in BNIP3 and NIX protein levels. Indeed, treatment of both wild-type and *PPTC7* KO cells with the iron chelator DFO induced similar fold changes in *BNIP3* (Fig 1E) and *NIX* (Fig 1G) mRNA transcripts across genotypes. At the protein level, however, expression of BNIP3 (Fig 1F) and NIX (Fig 1H) were substantially higher in DFO-treated *PPTC7* KO cells relative to other tested conditions. Notably, basal protein expression of BNIP3 and NIX in untreated *PPTC7* KO cells exceeded BNIP3 and NIX induction in DFO-treated wild-type cells (Fig 1F and H), indicating the magnitude of BNIP3 and NIX upregulation in the absence of PPTC7. These experiments further indicate that neither BNIP3 nor NIX is maximally upregulated in *PPTC7* KO cells in basal conditions.

The additive increase in BNIP3 and NIX expression in *PPTC7* KO cells upon DFO treatment led us to hypothesize that mitophagic flux may also be additive in DFO-treated *PPTC7* KO cells. To test this, we assayed wild-type and *Pptc7* KO MEFs expressing mt-Keima, a pH-sensitive fluorescent mitophagy sensor (Sun et al, 2017) using flow cytometry. We first exposed wild-type or *Pptc7* KO MEFs to varying concentrations of DFO and found that only the highest tested dose of DFO, 100 μM, induced mitophagy in wild-type cells (Fig S1D–F). Notably, the percentage of cells undergoing high mitophagic flux in wild-type cells treated with 100 μM DFO remained below the rates of mitophagic induction in untreated *Pptc7* KO cells (Fig S1D–F). We repeated these experiments in wild-type, *Pptc7* KO, and *Pptc7*/ *Bnip3*/*Bnip3l* triple KO (TKO) MEFs and found that 100 μM DFO induced an approximate threefold increase in mitophagic flux in both wild-type and *Pptc7* KO cells (Fig 1I–K). However, the absolute levels of mitophagy in DFO-treated *Pptc7* KO cells significantly exceeded those of DFO-treated wild-type cells as well as untreated *Pptc7* KO cells, with close to 80% of DFO-treated *Pptc7* KO cells undergoing high mitophagy (Fig 1I–K). Importantly, mt-Keima-positive *Pptc7*/*Bnip3*/*Nix* TKO cells failed to undergo appreciable mitophagy in the presence of 100 μM DFO, demonstrating their necessity in increasing mitophagic flux in *Pptc7* KO cells (Fig 1I–K). Collectively, these data demonstrate that BNIP3 and NIX are post-transcriptionally upregulated to induce mitophagy in *PPTC7* KO cells, and that transcriptional activation of HIF-1α can substantially enhance BNIP3/NIX protein expression and mitophagy in the absence of PPTC7.

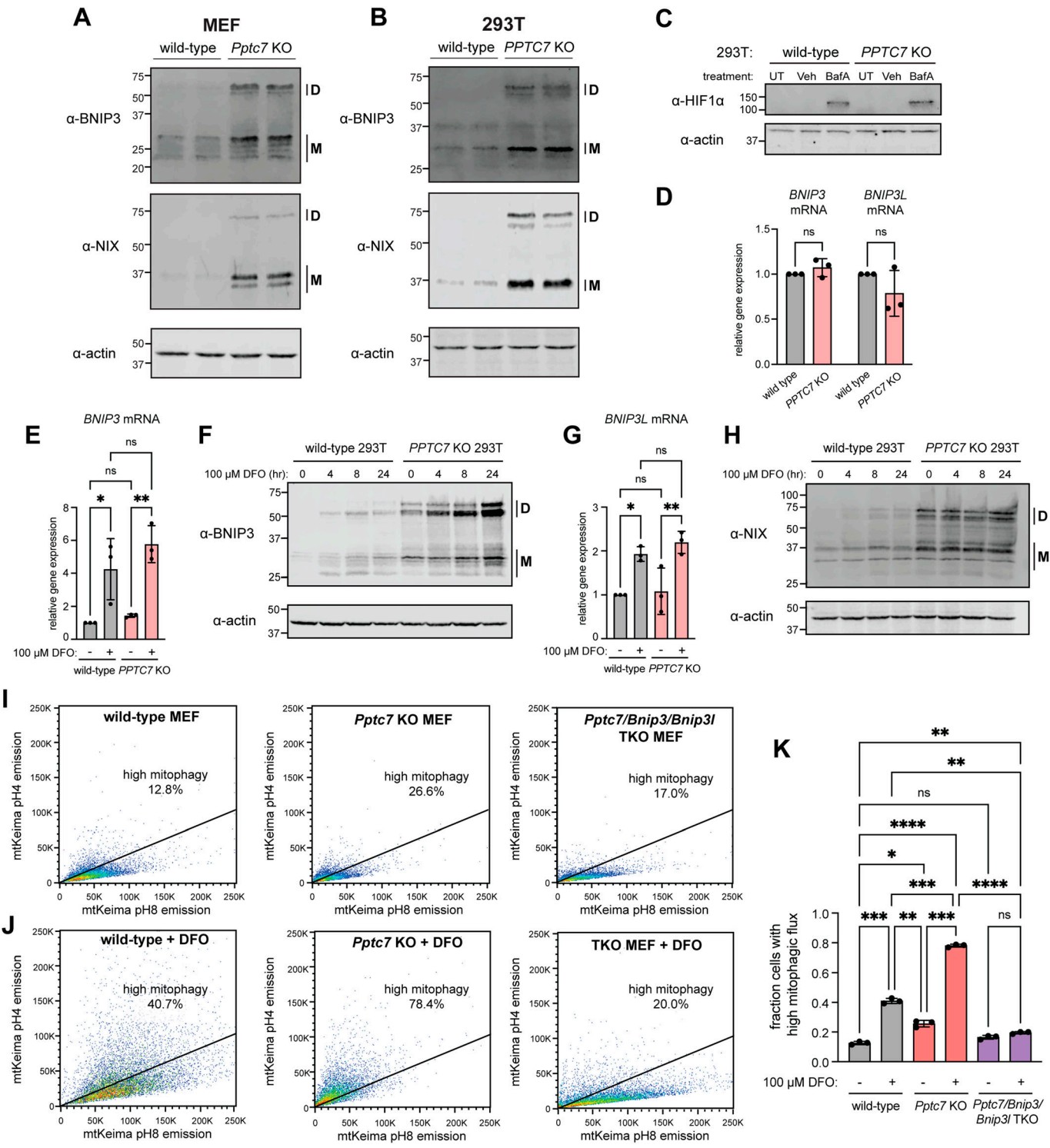

**Figure 1. BNIP3 and NIX are upregulated post-transcriptionally and independent of HIF-1α in *PPTC7* KO cells.**
**(A, B)** Western blots of BNIP3 (top panels), NIX (middle panels) and actin (serving as a load control, bottom panels) in wild-type or *Pptc7* KO MEFs (A) or in wild-type or *PPTC7* KO 293T cells (B). "D" indicates dimer species; "M" indicates monomer species. **(C)** Western blot for HIF-1α in untreated (UT), vehicle only (veh, 0.2% DMSO) or 100 nM bafilomycin A (BafA) for 16 h in wild-type and *PPTC7* KO 293T cells. Actin shown as a loading control. **(D)** qRT-PCR of *BNIP3* and *BNIP3L* (gene name of NIX) endogenous mRNA levels in wild-type (gray) and *PPTC7* KO (pink) 293T cells. Error bars represent SD; data points represent independent experiments. **(E)** qRT-PCR of *BNIP3* RNA levels in untreated and DFO-treated (100 μM, 24 h) wild-type (gray) and *PPTC7* KO (pink) 293T cells. \*\*P < 0.01, \*P < 0.05, ns = not significant, ordinary one-way ANOVA. Error bars represent SD; data points represent independent experiments. **(F)** Western blotting for endogenous BNIP3 levels in wild-type or *PPTC7* KO 293T cells treated with 100 μM DFO for indicated times. Actin shown as a loading control. **(G)** qRT-PCR of *BNIP3L* RNA levels in untreated and DFO-treated (100 μM, 24 h) wild-type (gray) and *PPTC7* KO (pink) 293T cells. \*\*P < 0.01, \*P < 0.05, ns = not significant, ordinary one-way ANOVA. Error bars represent SD; data points represent independent experiments. **(H)** Western

## PPTC7 enables BNIP3 and NIX turnover through proteasomal degradation

The elevated protein expression of BNIP3 and NIX in *PPTC7* KO cells implies that PPTC7 alters the synthesis or turnover rates of these mitophagy receptors. BNIP3 and NIX turnover has emerged as a critical regulatory step in limiting basal mitophagy, as evidenced by recent studies on the E3 ubiquitin ligase FBXL4 (Cao et al, 2023; Elcocks et al, 2023; Nguyen-Dien et al, 2023). Loss of *Fbxl4* phenotypically mirrors *Pptc7* KO in mice (Niemi et al, 2019; Alsina et al, 2020), and *PPTC7* and *FBXL4* have significant and positively correlated essentiality profiles across over one-thousand cancer cell lines (Fig S2A, [Dempster et al, 2019 *Preprint*]). These data suggest that PPTC7 and FBXL4 influence BNIP3 and NIX via similar mechanisms, leading us to hypothesize that BNIP3 and NIX have decreased turnover rates in *PPTC7* KO cells relative to wild-type cells.

To test this, we first sought to identify an experimental condition in which we could quantify endogenous BNIP3 and NIX turnover. We noted that DFO-mediated iron chelation was previously shown to decrease the protein level of select mitochondrial proteins in a manner that was reversible upon compound washout (Rensvold et al, 2013). As BNIP3 and NIX accumulate in response to DFO (Fig 1), we hypothesized that washout of DFO would induce BNIP3 turnover because of its short half-life (Schäfer et al, 2022) (Fig 2A). Indeed, treatment of wild-type 293T cells with DFO increased BNIP3 levels in a time-dependent manner, which returned to at or near baseline (i.e., untreated) levels 24 h after DFO washout (Fig 2B). Thus, DFO washout constitutes an experimental system in which we could test the effects of PPTC7 on the turnover of endogenous BNIP3 and NIX. We repeated these experiments in wild-type and *PPTC7* KO 293T cells and found that BNIP3 and NIX exhibited blunted turnover in *PPTC7* KO cells (Figs 2C–F and S2B–E). Modeling of BNIP3 and NIX decay rates showed that loss of PPTC7 extends the half-life of monomeric and dimeric populations of BNIP3 and NIX (Fig 2C–F). Whereas the dimer populations of BNIP3 and NIX have at least a doubling of protein half-life in *PPTC7* KO cells relative to wild-type cells, the half-lives of the monomeric populations of each mitophagy receptor could not be effectively modeled in *PPTC7* KO cells, consistent with substantial suppression of the turnover of these species of BNIP3 and NIX in the absence of PPTC7 (Fig 2C–F).

Given the slowed rates of BNIP3 and NIX turnover in *PPTC7* KO cells, we sought to understand the pathway(s) contributing to their degradation. The phenotypic similarities between loss-of-function models of *PPTC7* and *FBXL4*, as described above, suggest that these two proteins function in a similar pathway. In addition, previous work has shown that BNIP3 accumulates in response to the proteasomal inhibitor MG-132 (Park et al, 2013; Poole et al, 2021). If KO of *PPTC7* suppresses the proteasomal degradation of BNIP3 and NIX, we hypothesized that *PPTC7* KO cells would be less responsive to the proteasomal inhibitor MG-132 than matched wild-type cells. We

found that BNIP3 and NIX levels increased in wild-type cells upon MG-132 treatment (Fig 2G and H) but that the level of each receptor was significantly less responsive to MG-132 in *PPTC7* KO cells (Fig 2G and H). To further test this model, we exploited the DFO washout assay, predicting that if BNIP3 and NIX were turned over by proteasomal degradation upon DFO washout, MG-132 treatment would slow their turnover rates in wild-type cells but have a diminished effect in *PPTC7* KO cells. We found that, upon DFO washout, MG-132 treatment caused BNIP3 and NIX accumulation in wild-type cells, whereas the levels of these proteins remained largely unchanged in *PPTC7* KO cells under identical conditions (Fig 2I). Overall, these data demonstrate that PPTC7 enables the efficient turnover of BNIP3 and NIX in a manner that largely depends upon proteasomal degradation.

## PPTC7 requires an intact active site but not a mitochondrial targeting sequence to limit BNIP3 and NIX accumulation

As loss of *PPTC7* increases BNIP3 expression, we hypothesized that PPTC7 overexpression may diminish pseudohypoxia-induced BNIP3 upregulation. To test this, we overexpressed either cytosolic GFP or PPTC7-GFP in HeLa cells that were treated with the pseudohypoxia inducer cobalt chloride ($CoCl_2$). We fixed and immunolabeled these cells to examine endogenous BNIP3 levels relative to the general mitochondrial marker TOMM20. $CoCl_2$ robustly upregulated BNIP3 protein expression, and mitochondrial BNIP3 staining could be detected in nearly all cells transfected with cytosolic GFP (Fig 3A). In contrast to cytosolic GFP, PPTC7-GFP co-localized with the mitochondrial marker TOMM20 as expected (Fig 3A). Notably, mitochondrial BNIP3 signal was rarely detected in the subset of cells expressing PPTC7-GFP (Fig 3A and B). Similarly, overexpression of PPTC7 suppressed $CoCl_2$-induced mitophagic flux relative to identically treated wild-type cells expressing mt-Keima (Fig S3A and B). Finally, overexpression of PPTC7 in *Pptc7* KO MEFs rescued basal BNIP3 protein expression to levels seen in wild-type MEFs (Fig S3C). Collectively, these data show that overexpressed PPTC7 limits BNIP3 protein expression and mitophagy induction in both wild-type and *Pptc7* KO cells.

As these experiments indicated that overexpressed PPTC7 was properly localized and functional, we generated a series of mutants to understand the mechanism by which PPTC7 limits BNIP3 and NIX accumulation. PPTC7 consists of a PP2C phosphatase domain that is preceded by a short disordered N-terminus (Fig 3C), which includes a mitochondrial targeting sequence (i.e., MTS) that is processed at amino acid 14 (Calvo et al, 2017). Interestingly, when western blotting for overexpressed PPTC7, we found that the protein ran as a doublet, whereas a ΔMTS-PPTC7 mutant ran at the same molecular weight as the bottom band (Fig 3D), suggesting that wild-type PPTC7 expressed as both a full-length and processed form. Given that mitochondrial proteins are often processed upon their entrance to

---

blotting for endogenous NIX levels in wild-type or *PPTC7* KO 293T cells treated with 100 μM DFO for indicated times. Actin shown as a loading control. **(I)** FACS plots of basal mitophagy in wild-type (left), *Pptc7* KO (middle), and *Pptc7/Bnip3/Bnip3l* TKO MEFs using the mt-Keima fluorescence assay. Cells undergoing high mitophagy are above the diagonal line; percentages indicated in figure. **(J)** FACS plots of mitophagy rates upon 24 h of 100 μM DFO treatment in wild-type (left), *Pptc7* KO (middle), and *Pptc7/Bnip3/Bnip3l* TKO MEFs using the mt-Keima fluorescence assay. **(I, J, K)** Quantification of mt-Keima data shown in (I, J). ****$P < 0.0001$, ***$P < 0.001$, **$P < 0.01$, *$P < 0.05$, ns = not significant, ordinary one-way ANOVA. Error bars represent SD; data points represent individual biological replicates.

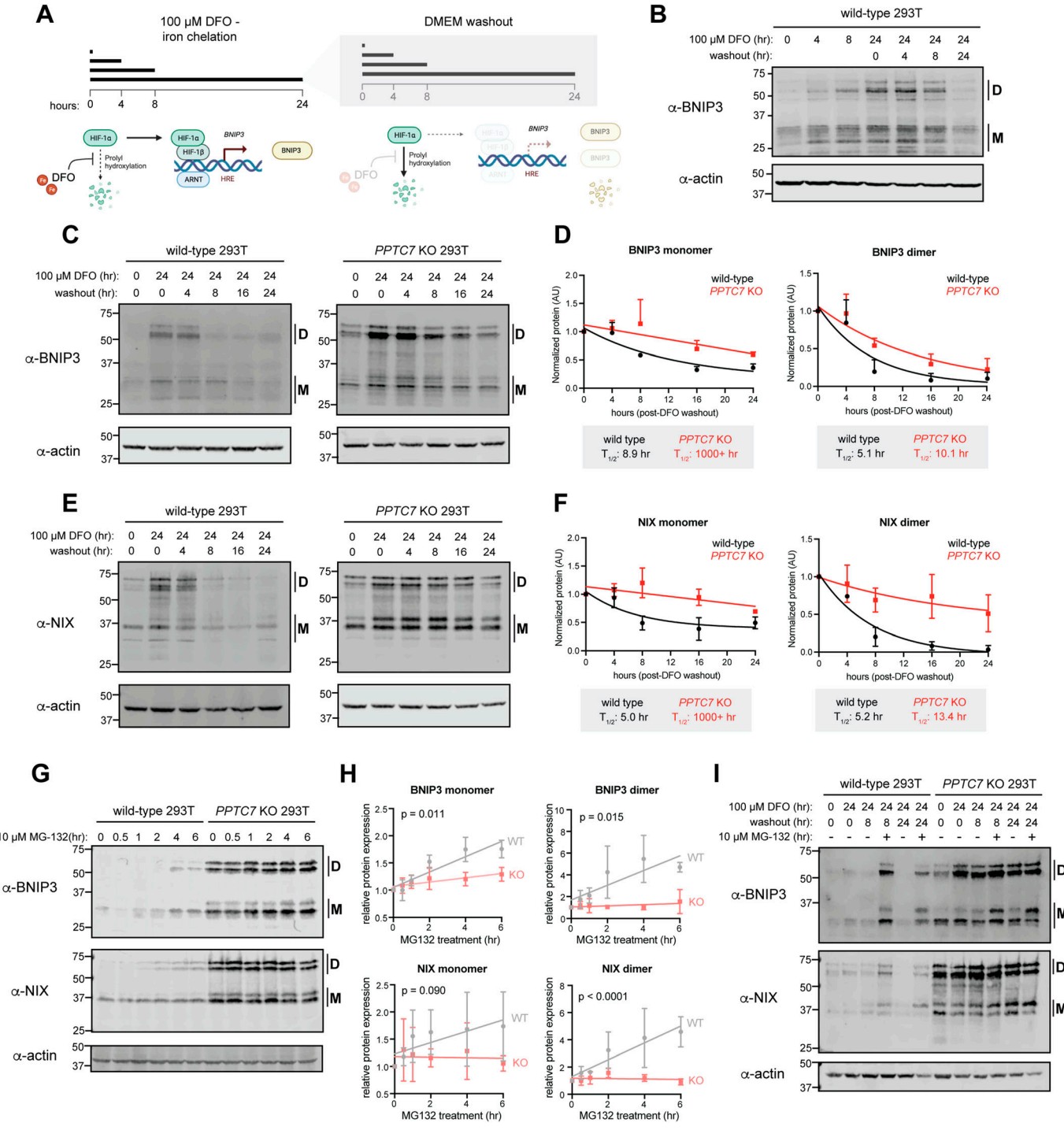

**Figure 2. BNIP3 and NIX have decreased turnover rates and are less responsive to proteasomal inhibition in *PPTC7* KO cells.**
**(A)** Schematic of DFO treatment and washout timeline and mechanism; figure made with BioRender. **(B)** Western blot of endogenous BNIP3 protein after indicated times of DFO treatment and washout, when applicable. "D" indicates dimer species; "M" indicates monomer species. Actin shown as a loading control. **(C)** Western blot of endogenous BNIP3 protein after indicated times of DFO treatment and washout in wild-type (left panel) and *PPTC7* KO (right panel) 293T cells. "D" indicates dimer species; "M" indicates monomer species. Actin shown as loading control. **(C, D)** Quantification of data shown in (C). BNIP3 monomer (left graph) or dimer (right graph) bands were quantified using densitometry, averaged, and plotted over time. Data were fit with a one-phase decay model to calculate protein half-lives (T$_{1/2}$), which are shown below each graph. Error bars represent standard deviations of normalized densitometry across three independent experiments. **(E)** Western blot of endogenous NIX protein after indicated times of DFO treatment and washout in wild-type (left panel) and *PPTC7* KO (right panel) 293T cells. "D" indicates dimer species; "M" indicates monomer species. Actin shown as loading control. **(E, F)** Quantification of data shown in (E). NIX monomer (left graph) or dimer (right graph) bands were quantified using densitometry, averaged, and plotted over time. Data were fit with a one-phase decay model to calculate protein half-lives (T$_{1/2}$), which are shown below each graph. Error bars represent standard deviations of normalized densitometry across three independent experiments. **(G)** Western blots of endogenous BNIP3 (top panel) and NIX (bottom panel) in wild-type and *PPTC7* KO 293T cells upon

the mitochondrial matrix, these data suggested that PPTC7 may reside in two locations, with the full-length isoform localizing outside of mitochondria, and the processed band within the matrix. To test this, we performed a protease protection assay and found that the top band of the PPTC7 doublet was susceptible to digestion mediated by proteinase K, whereas the bottom band was protected, consistent with dual localization (Fig 3E). Given these molecular insights, we used this overexpression system to test the necessity of PPTC7 catalytic activity and mitochondrial targeting in suppressing BNIP3 and NIX induction under pseudohypoxia.

We first mutated the PP2C phosphatase domain of PPTC7 at a key catalytic residue, D78, to alanine. We previously demonstrated that recombinant PPTC7 D78A was unable to dephosphorylate BNIP3 and NIX on mitochondria isolated from *Pptc7* KO MEFs (Niemi et al, 2023). Consistently, recombinant wild-type PPTC7, but not D78A PPTC7, caused a collapse in the laddering pattern of monomeric BNIP3 in mitochondria isolated from wild-type and *PPTC7* KO 293T cells (Fig S3D). These data show that not only that PPTC7 D78A lacks catalytic activity, but also that the upper monomeric bands seen on BNIP3 western blots represent phosphorylated intermediates that can be directly dephosphorylated by PPTC7. As such, we predicted that a D78A mutant would be unable to suppress BNIP3 and NIX accumulation during pseudohypoxia. We overexpressed wild-type and D78A PPTC7 in HeLa FLP-IN TREx cells and found that both constructs were doxycycline-induced to similar extents and, interestingly, also expressed at higher levels in the presence of $CoCl_2$, similar to BNIP3 and NIX (Fig 3F). Whereas overexpression of wild-type PPTC7 decreased BNIP3 and NIX accumulation in response to $CoCl_2$ treatment, the D78A mutant failed to suppress the induction of these mitophagy receptors (Fig 3F), indicating that PPTC7 requires an intact catalytic motif to influence BNIP3 and NIX protein expression. Importantly, however, recent structural modeling efforts suggest that BNIP3 and NIX may interact with PPTC7 proximal to its catalytic motif (Nguyen-Dien et al, 2024 *Preprint*, Fig S3E). Consistently, we find that D78A significantly decreases, but does not fully ablate, the physical interaction between PPTC7 and NIX (Fig S3F), similar to these findings (Nguyen-Dien et al, 2024 *Preprint*).

Given the dual localization of PPTC7, we next examined whether disrupting its mitochondrial localization would affect its ability to suppress $CoCl_2$-induced BNIP3 and NIX levels. We found that ΔMTS-PPTC7 fully suppressed $CoCl_2$-mediated NIX accumulation and partially suppressed BNIP3 accumulation (Fig 3G). This rescue of BNIP3 and NIX expression is consistent with a model in which a non-targeted, cytosolic PPTC7 influences mitophagy at sufficient (i.e., overexpressed) levels, suggesting a population of PPTC7 outside of mitochondria could mediate BNIP3 and NIX stability. To directly test this, we artificially anchored PPTC7 to the outer mitochondrial membrane (OMM) by engineering a PPTC7 construct that both lacks an MTS and is fused to OMP25, a tail-anchored protein that targets to the OMM (Horie et al, 2002). Consistent with

our hypothesis, ΔMTS-PPTC7-OMP25 blunts the accumulation of BNIP3 and NIX under both basal and $CoCl_2$-treated conditions (Fig 3H), demonstrating that OMM-localized PPTC7 is sufficient to suppress BNIP3 and NIX protein expression. Collectively, these data show that PPTC7 requires an intact active site but not its mitochondrial targeting sequence to suppress BNIP3 and NIX expression in pseudohypoxic conditions, consistent with a role for PPTC7 in regulating these mitophagy receptors outside of mitochondria.

## PPTC7 proximally and dynamically interacts with BNIP3 and NIX in cells

Our data indicate that a pool of PPTC7 resides outside of mitochondria, which may interact with BNIP3 and NIX to promote their proteasomal turnover. We sought to test this model by determining whether BNIP3 and NIX interact with PPTC7 in cells through miniTurbo-based proximity labeling experiments. We expressed PPTC7-V5-miniTurbo, as well as two control constructs, in 293T cells with or without DFO treatment. We treated half of the samples with exogenous biotin (which facilitates miniTurbo-based proximity labeling) and left the remaining cells untreated to control for potential non-specific interactions. After biotinylation, we lysed cells, performed a pulldown with streptavidin beads to enrich for biotinylated proximal interactors, and probed for interaction with BNIP3 and NIX via Western blot. Whereas PPTC7-V5-miniTurbo expressed equally across all conditions, the streptavidin pull-down revealed interactions with BNIP3 and NIX only in the biotin-treated samples, demonstrating specific proximal labeling (Fig 4A). Importantly, these interactions were also specific to PPTC7-V5-miniTurbo and were not seen in either vector-only or V5-miniTurbo control samples (Fig 4A). Furthermore, our western blotting approach resolves both dimer and monomer forms of BNIP3 and NIX, revealing that both forms of these mitophagy receptors proximally associate with PPTC7-V5-miniTurbo. These data, combined with our data demonstrating that recombinant PPTC7 can directly dephosphorylate BNIP3 in vitro (Fig S3D), strongly suggest that PPTC7 directly interacts with BNIP3 and NIX within the native cellular environment.

These data support a model in which PPTC7 is directly recruited to BNIP3 and NIX to promote their turnover. Our DFO washout experiments offer insights into the kinetics of this turnover, which allowed us to test whether the PPTC7-BNIP3/NIX interactions are dynamic throughout the turnover process. Our experiments show that BNIP3 is present at near peak levels 4 h post-DFO washout (Fig 2B and C), leading us to hypothesize that PPTC7 would have enhanced recruitment to BNIP3 and NIX during acute DFO washout to facilitate the turnover of BNIP3 and NIX. To test this, we repeated proximity labeling of PPTC7-V5-miniTurbo under conditions of acute (i.e., 4 h) DFO washout. Interestingly, whereas the PPTC7-BNIP3/NIX interactions are enriched upon 24 h of DFO treatment,

---

treatment with 10 μM MG-132 for the indicated timeframes. "D" indicates dimer species; "M" indicates monomer species. Actin shown as a loading control. **(G, H)** Quantification of BNIP3 and NIX monomer (left graphs) and dimer (right graphs) populations in wild-type (gray) and *PPTC7* KO (pink) cells shown in (G). Bands were quantified using densitometry, averaged, and plotted over time. Data were analyzed via linear regression, and the significance between slopes was calculated using Analysis of Covariance (ANCOVA). **(I)** Western blot of endogenous BNIP3 and NIX proteins in wild-type and *PPTC7* KO cells after DFO treatment and subsequent washout in the presence or absence of 10 μM MG-132. "D" indicates dimer species; "M" indicates monomer species. Actin shown as a loading control.

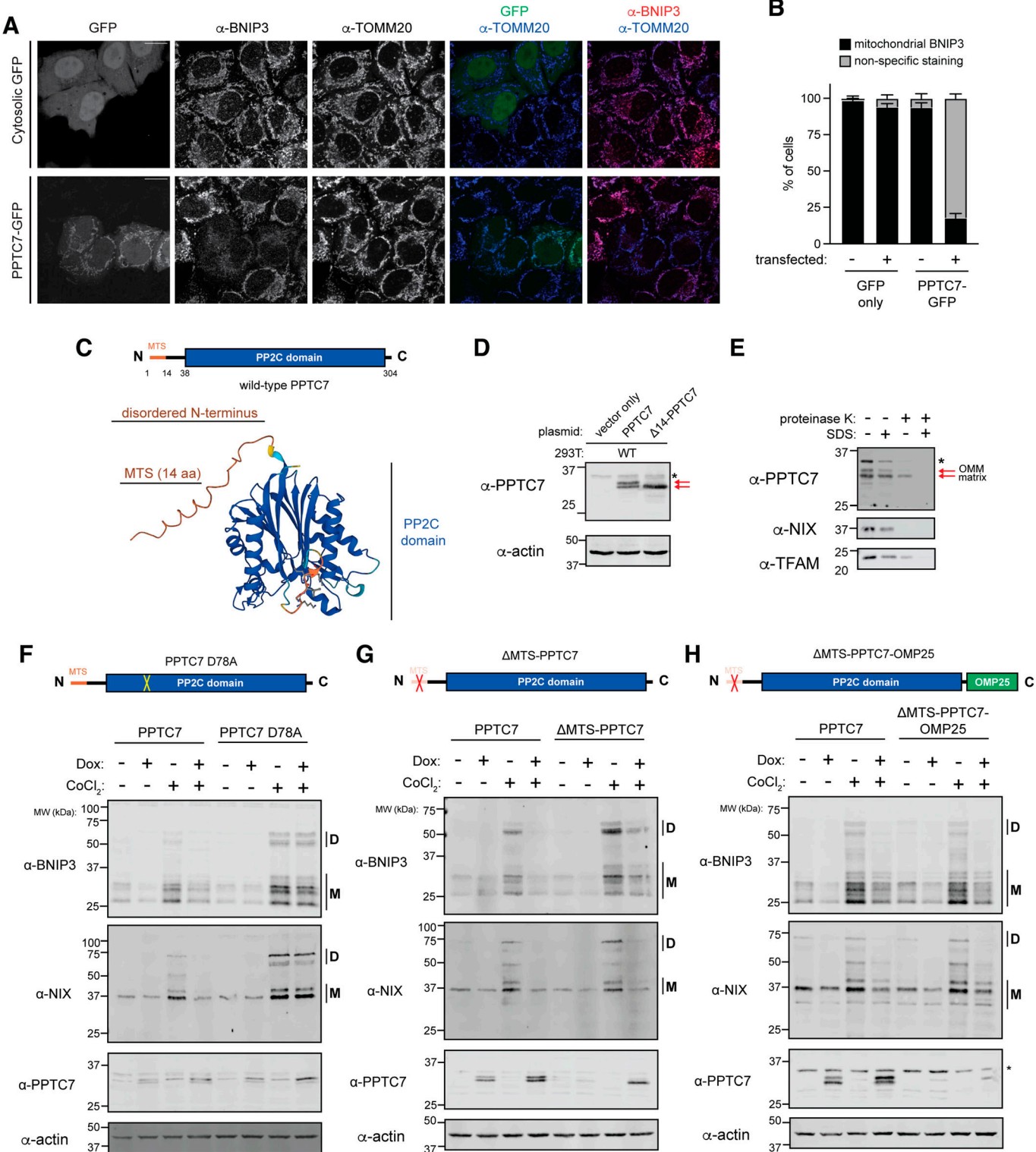

**Figure 3. PPTC7 requires an intact catalytic motif but not mitochondrial targeting to suppress BNIP3 and NIX accumulation.**
**(A)** Representative single-plane confocal images of GFP only (top panels) or PPTC7-GFP (bottom panels) expressed in HeLa cells treated for 12 h with 500 μM CoCl₂. Cells were stained for BNIP3 (second column) or TOMM20 (third column). Overlays are shown for GFP and TOMM20 (fourth column) and GFP and BNIP3 (fifth column). Scale bar = 20 μm. **(A, B)** Quantification of data shown in (A), mitochondrial BNIP3 staining versus non-specific staining in cells overexpressing GFP only or PPTC7-GFP versus matched untransfected controls for each experiment. Error bars represent the standard error of the mean of three independent experiments. **(C)** Schematic of PPTC7 features, including a mitochondrial targeting sequence (MTS) and PP2C phosphatase domain, top. Bottom, AlphaFold2 representation of PPTC7 structure; predicted disordered N-terminus and PP2C phosphatase domains indicated. **(D)** Western blot of 293T cells overexpressing PPTC7 or a ΔMTS-PPTC7 mutant. Red arrows indicate dual species in

similar to our previous experiment, they are further enhanced upon acute 4-h DFO washout (Fig 4B). These data demonstrate that PPTC7 is dynamically recruited to BNIP3 and NIX to promote their turnover upon resolution of DFO-mediated pseudohypoxia.

To further explore the dynamic nature of the PPTC7-BNIP3 interaction, we exploited the recent observation that BNIP3 enriches at LC3-positive punctate structures which likely represent nascent mitophagosomes (Gok et al, 2023). We hypothesized that PPTC7 would also enrich at these punctate structures under conditions of pseudohypoxia, and that this localization would be enriched upon resolution of pseudohypoxia. We overexpressed PPTC7-GFP in U2OS cells treated with the iron chelator DFP for 24 h, and then fixed and immunolabeled for BNIP3 and TOMM20. Examination of over 400 BNIP3-enriched foci in DFP-treated cells revealed that ~14% of these structures were co-enriched for PPTC7-GFP (Fig 4C). Importantly, these foci were co-localized with the mitochondrial marker TOMM20 (Fig 4C), demonstrating these interactions occur at mitochondria and not other organelles, such the ER, where BNIP3 has been reported to localize (Zhang et al, 2009; Hanna et al, 2012). Remarkably, we found that PPTC7-GFP showed almost a threefold increase co-enrichment (~46%) with BNIP3-enriched foci 4 h after DFP washout relative to DFP treatment alone (Fig 4D). These data, along with our proximity labeling experiments, suggest that PPTC7 is actively recruited to BNIP3 and NIX under iron chelation conditions that promote their turnover. Collectively, our data are consistent with a model in which a pool of PPTC7 dynamically localizes outside of mitochondria to associate with BNIP3 and NIX–likely through a direct interaction–to promote their ubiquitin-mediated turnover.

## Discussion

PPTC7 is one of twelve phosphatases that localize to mammalian mitochondria (Niemi & Pagliarini, 2021). Conserved through budding yeast (where it is named Ptc7p), PPTC7 has been linked to the maintenance of metabolism and mitochondrial homeostasis across organisms (Martín-Montalvo et al, 2013; Guo et al, 2017a, 2017b; Gonzalez-Mariscal et al, 2017; González-Mariscal et al, 2018; Niemi et al, 2019, 2023). Whereas we previously have performed phosphoproteomics to identify candidate substrates of Ptc7p (Guo et al, 2017a, 2017b) and PPTC7 (Niemi et al, 2019, 2023), the precise roles of these phosphatases in regulating mitochondrial homeostasis remain unclear. In this study, we began to illuminate important aspects underlying the regulation and function of PPTC7 and how this phosphatase influences BNIP3- and NIX-mediated mitophagy.

The data presented in this study demonstrate that BNIP3 and NIX turnover is a tightly regulated and highly dynamic process. Using a model of DFO-mediated iron chelation followed by compound washout, we showed that endogenous BNIP3 and NIX were rapidly turned over in wild-type cells in a manner that was slowed by proteasomal inhibition. Our data further showed that BNIP3 and NIX exhibit longer half-lives and are less responsive to proteasomal inhibition in *PPTC7* KO cells, consistent with a model in which the phosphatase functions to promote the ubiquitin-mediated turnover of these mitophagy receptors. Indeed, two studies published during the preparation of this article showed that PPTC7 coordinates BNIP3 and NIX degradation by acting as a molecular scaffold for the E3 ligase FBXL4 at the OMM (Nguyen-Dien et al, 2024 *Preprint*; Sun et al, 2024). Consistently, we found that PPTC7 is dual-localized and that its anchoring on the OMM decreased BNIP3 and NIX protein levels under pseudohypoxia. Furthermore, we found that PPTC7 associated with BNIP3 and NIX via proximity labeling experiments, demonstrating a likely direct interaction between the phosphatase and these mitophagy receptors in cells. These data indicate that PPTC7 has critical functions at the OMM to regulate BNIP3- and NIX-mediated mitophagy.

The apparent dual functionality of PPTC7 across mitochondrial compartments leads to interesting questions regarding its regulation. Some insights into this regulation may be gleaned from its yeast ortholog Ptc7p, which is also dual localized (Juneau et al, 2009). Yeast *PTC7* is alternative spliced, with the full-length gene containing an in-frame intron that encodes a transmembrane domain that anchors it within the endoplasmic reticulum and/or nuclear envelope (Juneau et al, 2009; Williams et al, 2014), where it performs as yet unknown functions. However, upon select stimuli, such as nutrient switching to a non-fermentable carbon source, *PTC7* is spliced to remove its TM domain, generating a mitochondrial targeting sequence to promote its import to the matrix (Juneau et al, 2009). These data suggest that Ptc7p and PPTC7 may have evolved distinct but important regulatory mechanisms to regulate their entrance into the mitochondrial matrix.

Our data also indicate that PPTC7 is dynamically recruited to the OMM, as the proximal interactions between PPTC7 and BNIP3/NIX were enriched upon resolution of pseudohypoxia (i.e., the washout of iron chelator). This conclusion is further supported by the enhanced co-localization of fluorescently tagged PPTC7 with BNIP3-positive punctae in conditions of iron chelator washout, as well as a protein-level increase in the full length, OMM PPTC7 isoform during pseudohypoxia (Fig 3F–H). Furthermore, our data suggest a precisely coordinated spatial organization of BNIP3 and PPTC7 at foci that likely represent mitophagosome formation sites (Gok et al, 2023). Interestingly, TMEM11 was recently found to co-localize with BNIP3 and NIX at mitophagic punctae, and its KO increases both BNIP3-positive foci as well as mitophagic flux (Gok et al, 2023). These data suggest that PPTC7 and TMEM11 may function in similar

---

wild-type PPTC7 expression. * represents a non-specific band. Actin shown as a loading control. **(E)** PPTC7 protease protection assay. Mitochondria from HeLa FLP-IN cells expressing PPTC7 were isolated and treated with SDS, proteinase K, or both and resolved via SDS–PAGE. Western blots of PPTC7 (top), the outer mitochondrial membrane marker NIX (middle) and the matrix marker TFAM (bottom) shown. **(F, G, H)** Western blots of BNIP3, NIX, and PPTC7 depicting the ability of various mutants of PPTC7 to suppress BNIP3 and NIX accumulation in response to $CoCl_2$ treatment. "D" indicates dimer species, "M" indicates monomer species. In (F), a catalytic mutant of PPTC7, D78A, cannot effectively suppress BNIP3 and NIX accumulation relative to wild-type PPTC7. In (G), the ΔMTS-PPTC7 mutant partially or fully suppresses BNIP3 and NIX accumulation, respectively. In (H), a mutant that artificially anchors PPTC7 to the outer mitochondrial membrane, ΔMTS-PPTC7-OMP25, suppresses BNIP3 and NIX accumulation. Actin shown as a loading control.

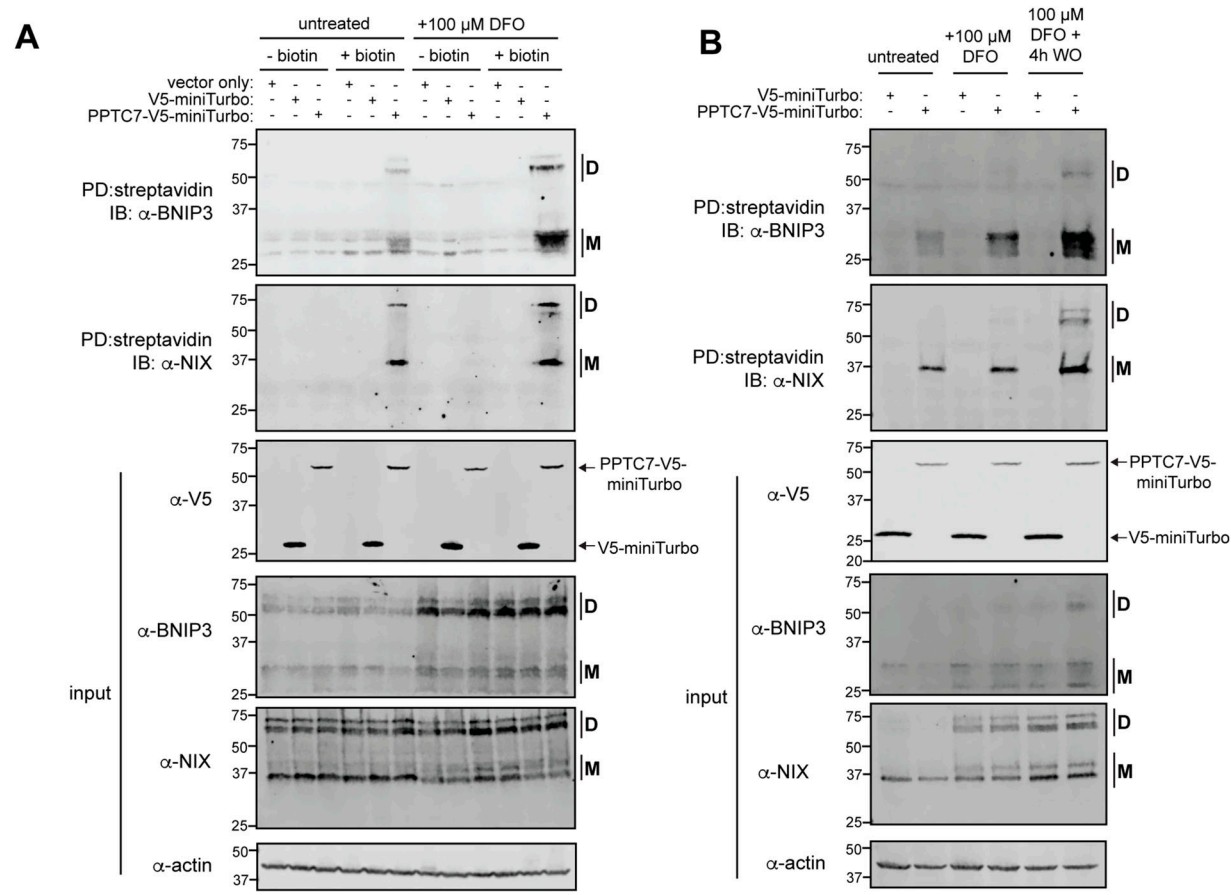

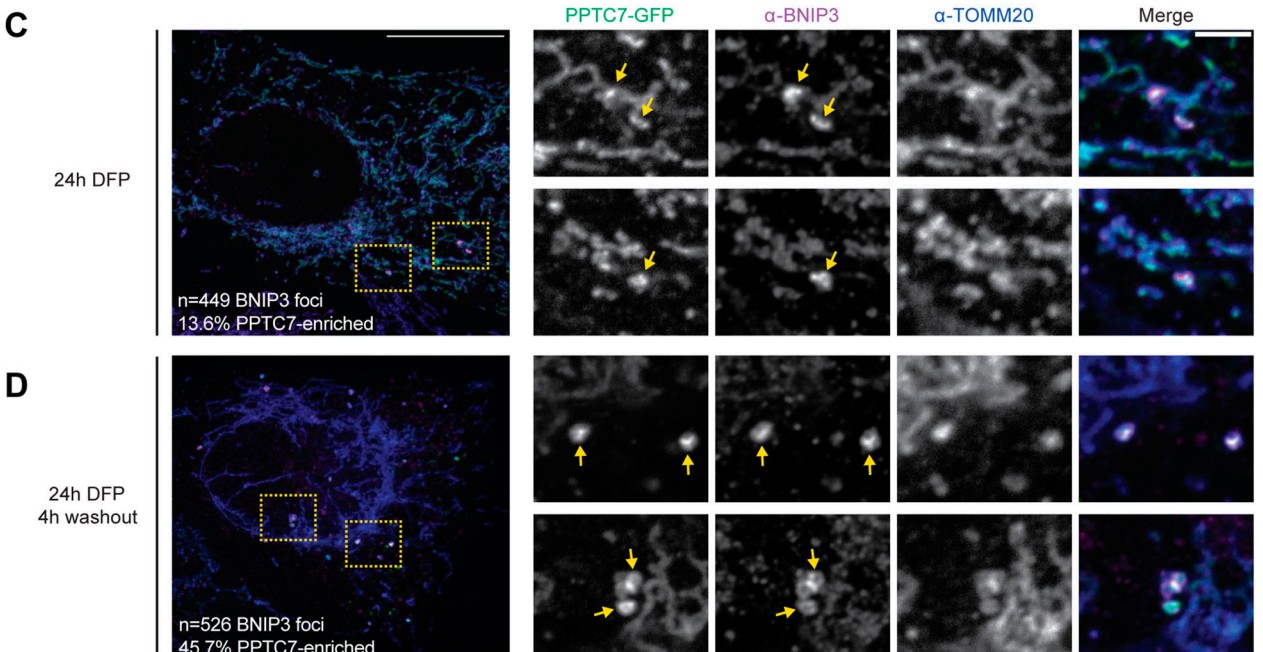

**Figure 4. PPTC7 proximally and dynamically interacts with BNIP3 and NIX in cells.**
**(A)** Proximity labeling of PPTC7-V5-miniTurbo in 293T cells with or without 24-h deferoxamine (DFO) treatment. PPTC7-V5-miniTurbo, as well as vector only or V5-miniTurbo-only controls, were transfected into 293T cells. Streptavidin pulldowns were used to enrich for PPTC7-V5-miniTurbo interactors, which were run on SDS–PAGE

complexes to regulate BNIP3- and NIX-mediated turnover and/or mitophagy. Whether these complexes include FBXL4 and how they may actively remodel to promote or limit mitophagy are key questions that warrant further investigation.

Previous studies of *Fbxl4* and *Pptc7* KO mouse models showed that each display similar pathophysiological profiles, including metabolic dysfunction, broad loss of mitochondrial protein levels, and perinatal lethality (Niemi et al, 2019; Alsina et al, 2020). Importantly, KO of *Bnip3* and *Nix* alleviated many of these phenotypes in cells (Cao et al, 2023; Elcocks et al, 2023; Nguyen-Dien et al, 2023; Niemi et al, 2023) and mice (Cao et al, 2023; Sun et al, 2024), suggesting that loss of *Fbxl4* or *Pptc7* drives excessive BNIP3- and NIX-mediated mitophagy. Despite these advances, the precise mechanism by which PPTC7 influences BNIP3 and NIX protein levels remains unclear. We previously found that BNIP3 and NIX are hyperphosphorylated in *PPTC7* KO cells and tissues, suggesting PPTC7 may influence mitophagy via their dephosphorylation (Niemi et al, 2019, 2023). Indeed, phosphorylation of BNIP3 or NIX can enhance their stability or their ability to induce mitophagy (Zhu et al, 2013; Rogov et al, 2017; Poole et al, 2021; He et al, 2022), suggesting dephosphorylation may be required to suppress their activity. However, two recent studies suggest that PPTC7 promotes BNIP3 and NIX degradation independent of its phosphatase activity (Nguyen-Dien et al, 2024 *Preprint*; Sun et al, 2024). Nguyen-Dien et al found that the mutant used in our study, PPTC7 D78A, binds BNIP3 and NIX less robustly than wild-type PPTC7—a finding which we confirmed. However, we found that a minor fraction of D78A immunoprecipitated NIX, and it is unclear whether compromised binding, a lack of phosphatase activity, or a combination contribute to the inability of PPTC7 D78A to promote BNIP3 and NIX turnover. To circumvent these confounding factors, Nguyen-Dien et al generated a mutant of PPTC7, D290N, with compromised phosphatase activity that still promotes BNIP3/NIX turnover (Nguyen-Dien et al, 2024 *Preprint*). It is unclear, however, if this mutant has residual phosphatase activity and whether this could contribute to the turnover of these mitophagy receptors. Further complexities are found across these studies, as Nguyen-Dien et al propose that PPTC7 bridges BNIP3/NIX directly to FBXL4, whereas Sun et al propose that PPTC7 binds to Cullin-Rbx1—important components within the SCF complex. Collectively, these data suggest a complex model in which PPTC7 mediates BNIP3/NIX degradation through interactions with diverse protein partners. Careful dissection of the mechanisms by which PPTC7 limits BNIP3- and NIX-mediated mitophagy, as well as the extent to which phosphorylation affects BNIP3 and NIX stability, should be active areas of investigation in the future.

Collectively, our data reveal an unexpected role for the mitochondrial protein phosphatase PPTC7 in the regulation of receptor-mediated mitophagy at the outer mitochondria membrane.

Whereas KO of *Bnip3* and *Bnip3l* largely rescue the decreases in mitochondrial protein levels present in *Pptc7* KO tissues and cells, their collective KO fails to fully rescue metabolic defects associated with loss of this phosphatase (Niemi et al, 2023). This, together with our data indicating PPTC7 exists in two distinct localization-dependent pools in cells, suggests that PPTC7 regulates additional mitochondrial functions within the matrix. It is possible that PPTC7 mediates such functions by influencing the mitophagic selectivity of matrix-localized substrates, as occurs in yeast (Tal et al, 2007; Abeliovich et al, 2013; Kolitsida et al, 2019, 2023). Alternatively, it is possible that PPTC7 maintains mitochondrial metabolism completely independent of its role in mitophagic signaling, potentially through the regulation of one or more candidate substrates previously identified via phosphoproteomic analyses (Niemi et al, 2019, 2023). Disentangling the matrix-mediated functions of PPTC7 versus those that regulate functions outside of mitochondria, and how these ultimately relate to mitophagic flux, will be a key next step in understanding the role of this phosphatase in modulating mitochondrial homeostasis.

# Materials and Methods

### Supplies and reagents

DFO and cobalt chloride were purchased from MilliporeSigma (Burlington, MA). MG-132 and polyethylenimine was purchased from Thermo Fisher Scientific. Lipofectamine 3000 was purchased from Thermo Fisher Scientific. Fugene 6 was purchased from Promega. Multiple plasmids were ordered or generated for this study, including: myc-*Mm*BNIP3-FL in pcDNA3.1, which was a kind gift from Joseph Gordon (#100796; Addgene, [Diehl-Jones et al, 2015]) and V5-miniTurbo-NES in pcDNA3.1, which was a kind gift from Alice Ting (#107170; Addgene, [Branon et al, 2018]). pCMV-SPORT6-*Mm*NIX was purchased from Horizon Discoveries. pcDNA3.1-PPTC7-FLAG, pcDNA3.1-FLAG-*Mm*NIX, pcDNA3.1-PPTC7-GFP, pcDNA3.1-PPTC7-OMP25-FLAG, pcDNA3.1-PPTC7-V5-miniTurbo, and pcDNA3.1-ΔMTS-PPTC7-FLAG were generated for this study through standard PCR and restriction-enzyme based cloning techniques. The ΔMTS-PPTC7 truncates the first 14 amino acids from the full-length construct. Plasmids for the HeLa FLP-IN TREx system were generated using Gateway cloning. Briefly, constructs were PCR amplified with primers containing attB1 or attB2 sequences. These fragments were incubated with pDONR221 (a kind gift from Julia Pagan) and BP clonase for recombination. Positive constructs were incubated with LR clonase and the pcDNA5/FRT/TO-Venus-Flag-Gateway destination vector, which was a kind gift from Jonathon Pines (#40999; Addgene, [Di Fiore & Pines, 2010]). All cloned plasmids were validated via Sanger sequencing.

---

gels and western blotted for BNIP3 (top blot) or NIX (second blot). Only PPTC7-V5-miniTurbo + biotin samples pulled down BNIP3 and NIX (lanes 6 and 12, streptavidin pull-down gels), indicating specific binding. Western blots shown for reaction input for pulldowns for V5 (showing miniTurbo constructs), BNIP3, NIX, and actin (serving as a load control). **(B)** Proximity labeling of PPTC7-V5-miniTurbo in 293T cells with after 24 h DFO treatment with or without 4 h DFO washout. Streptavidin pulldowns were used to enrich PPTC7-V5-miniTurbo interactors as described in (A). Western blots shown for reaction input for pulldowns as described in (A). **(C)** A representative maximum z-projection confocal image (left) and corresponding single plane insets (right) are shown of a U2OS cell overexpressing PPTC7-GFP and treated with deferiprone for 24 h. Cells were fixed and stained for BNIP3 and TOMM20 to visualize co-enrichment of PPTC7 with BNIP3-enriched foci (n = 449). **(D)** As in (C) for cells treated for 24 h with deferiprone and washed for an additional 4 h before fixation. Cells were stained to visualize co-enrichment of PPTC7 with BNIP3-enriched foci (n = 526).

## Cell culture and transfection

293T cells were acquired from the American Type Culture Collection. *PPTC7* KO 293T cells were generated as previously described (Meyer et al, 2020) and MEF cells were generated from wild-type and *Pptc7*<sup>−/−</sup> mice embryos as previously described (Niemi et al, 2023). Wild-type, *Pptc7* KO, and *Pptc7*/*Bnip3*/*Bnip3l* TKO MEFs were transduced with mt-Keima as previously described (Niemi et al, 2023). HeLa FLP-IN TREx cells stably expressing mt-Keima were a kind gift from Dr. Julia Pagan. Cells were cultured in growth media (DMEM supplemented with 10% heat-inactivated FBS and 1× penicillin/streptomycin). Cells were grown in a temperature-controlled incubator at 37°C and 5% $CO_2$. Transient plasmid transfection into 293T wild-type and *PPTC7* KO cells was performed with polyethylenimine for 24–48 h. Transient plasmid transfection into U2OS and HeLa cells was performed with Lipofectamine 3000 for 5 h. Stable plasmid transfection into HeLa FLP-IN TREx cells were performed in the presence of pOG44 in a ratio of 0.5 µg pcDNA5 to 2 µg pOGG44. Transfections were performed with Fugene 6 (Promega) per manufacturer's directions for 24–48 h before selection with 400 µg/ml hygromycin B.

## SDS–PAGE and immunoblotting

Cells were lysed with radioimmunoprecipitation buffer (RIPA; 0.5% wt/vol sodium deoxycholate, 150 mM sodium chloride, 1.0% vol/vol IGEPAL CA-630, 1.0% SDS, 50 mM Tris pH 8.0. 1 mM EDTA pH 8.0 in water) supplemented with 1x protease inhibitor cocktail (0.5 µg/ml pepstatin A, chymostatin, antipain, leupeptin, and aprotinin) and 1x phosphatase inhibitor cocktail (0.5 mM imidazole, 0.25 mM sodium fluoride, 0.3 mM sodium molybdate, 0.25 mM sodium orthovanadate, and 1 mM sodium tartrate) unless otherwise specified. After generating cell lysates, samples were clarified by centrifugation (21,100*g*) at 4°C for 10 min, snap frozen in liquid nitrogen, and stored at –80°C before use. All samples were quantified with the bicinchoninic acid assay kit (Thermo Fisher Scientific). Lysates were mixed with 5x sample buffer (312 mM Tris-Base, 25% wt/vol sucrose, 5% wt/vol SDS, 0.05% wt/vol bromophenol blue, 5% vol/vol *β*-mercaptoethanol, pH 6.8) and boiled at 95°C for 10 min. Lysates (20–40 µg) were run on SDS–PAGE gels with Precision All-Blue Protein Standards (Bio-Rad) before being transferred onto nitrocellulose membranes. Membranes were incubated with primary antibodies with 2% BSA or 3% nonfat diary milk in TBS-T for periods as indicated above. Primary antibodies used in immunoblotting include: anti-human BNIP3 (catalog #44060, dilution 1:1,000, 48 h incubation at 4°C; Cell Signaling Technology [CST]), anti-rodent BNIP3 (catalog #3769, dilution 1:1,000, 48 h incubation at 4°C; CST), anti-NIX (catalog #12396, dilution 1:1,000, 48 h incubation at 4°C; CST), anti-PPTC7 (catalog #NBP190654, dilution 1:1,000, 48 h incubation at 4°C; Novus), anti-HIF-1*α* (catalog #36169, dilution 1:1,000, overnight incubation at 4°C; CST), anti-*β*-actin (catalog #3700, dilution 1:1,000; CST; catalog #4970, dilution 1:1,000; CST; and catalog #ab170325, dilution 1:1,000; overnight incubation at 4°C; Abcam), anti-FLAG (catalog #F1804, dilution 1:2,000, overnight incubation at 4°C; Sigma-Aldrich), anti-V5 (catalog #PIMA515253, dilution 1:1,000, overnight incubation at 4°C; Thermo Fisher Scientific), and anti-myc (catalog #MA121316, dilution 1:1,000, overnight incubation at 4°C;

Thermo Fisher Scientific). Membranes were washed 2–3x with TBS-T for 5 min per wash and incubated with corresponding fluorophore-conjugated antibodies for 30 min at room temperature. Anti-680 or anti-800 conjugated mouse or rabbit antibodies (LiCOR) were used for detection. Membranes were then washed 2–3x with TBS-T for 5 min per wash and were imaged with a LiCOR OdysseyFC instrument using Image Studio software (LiCOR; version 5.2).

## Co-immunoprecipitation assay

HeLa FLP-IN TREx mt-Keima-positive cells exogenously expressing doxycycline-inducible FLAG-tagged PPTC7 WT or PPTC7 D78A were treated with 10 µM doxycycline for 48 h and $CoCl_2$ for 16 h before harvesting. Cell pellets were lysed in a Tris-Triton lysis buffer (50 mM Tris-Cl pH 7.5, 150 mM NaCl, 1 mM EDTA, 1 mM EGTA, 5 mM $MgCl_2$, 1 mM *β*-glycerophosphate and 1% Triton) containing a protease inhibitor cocktail (PI78437; Thermo Fisher Scientific) and phosphatase inhibitors (0.5 mM imidazole, 0.25 mM sodium fluoride, 0.3 mM sodium molybdate, 0.25 mM sodium orthovanadate, and 1 mM sodium tartrate) on ice for 30 min. Cell lysates were separated by centrifugation at 20,000*g* for 10 min at 4°C. To precipitate FLAG-tagged PPTC7 WT and D78A, cell lysates were incubated on a rotator for 2 h at 4°C with Anti-FLAG M2 Affinity Gel beads (A2220; Sigma-Aldrich). The bound species were washed with Tris-Triton lysis buffer four times, with 1-min centrifugations at 2,000*g* between each wash, before elution with 15 µl 2x sample buffer at 95°C for 10 min. The samples were then run on an SDS–PAGE gel and immunoblotted for NIX and FLAG.

## RNA extraction and qRT-PCR

For RNA extraction, the collected cell pellets were processed using Monarch RNA extraction kit (New England BioLabs). The RNA samples were then quantified and normalized before cDNA synthesis (Lambda Biotechnologies). Quantitative PCR analyses with SYBR Green PCR Master Mix (Applied Biosystems) were performed on a Bio-Rad CFX-96 Touch Real-Time PCR Detection System controlled by CFX Maestro (ver2.2) on a computer. Primers used were: GAPDH forward 5′-TTCGCTCTCTGCTCCTCCTGTT-3′, GAPDH reverse 5′-GCCCAATACGACCAAATCCGTTGA-3′, BNIP3 forward 5′-GCCCACCTCGCTCGCAGACAC-3′, BNIP3 reverse 5′-CAATCCGATGGCCAGCAAATGAGA-3′, NIX forward 5′-CTACCCATGAACAGCAGCAA-3′, and NIX reverse 5′-ATCTGCCCATCTTCTTGTGG-3′.

## Flow cytometry

For all experiments, wild-type MEFs or HeLa cells (i.e., with no mt-Keima expressed) were used as a negative control for gating and were plated at the same density as other cells in the corresponding experiment. Cells were grown in standard DMEM media (25 mM glucose, 2 mM glutamine, 10% heat-inactivated FBS, 1x penicillin/streptomycin) for 48 h. Cells were harvested by trypsinization and were resuspended in FluoroBrite media with 0.8% heat-inactivated FBS in 5 ml polystyrene round-bottom tubes (Falcon) immediately before the flow cytometry experiments. The LSR-Fortessa (BD Biosciences) flow cytometer was used and was controlled by BDFACSDiva software (version 9.0). Channels used include: FSC

(488 nm), SSC (excitation 488 nm, emission 488 nm), QDot 605 (excitation 405 nm, emission 610 nm), and PE-TexasRed (excitation 585 nm, emission 610 nm). The laser intensities for Qdot 605 and PE-TexasRed were changed based on the emission profile of wild-type MEF cells for each experiment and were kept constant throughout the experiment. Cells were gated to select for live cells, single cells, and mt-Keima positive cells sequentially. Once gates were established, they were used for the duration of that experiment. Flow cytometry data were processed with FlowJo (version 10.2.2), and the high mitophagy gate was drawn during analysis.

### Immunofluorescence assay and confocal microscopy

For immunofluorescence, cells grown on glass-bottom cover dishes (CellVis) were fixed in 4% PFA solution in PBS (15 min, room temperature). Fixed cells were permeabilized (0.1% Triton X-100 in PBS), blocked (10% FBS and 0.1% Triton X-100 in PBS), and then incubated overnight at 4°C with the indicated primary antibodies in blocking buffer. Primary antibodies used in immunofluorescence assays include anti-TOMM20 (catalog #56783, dilution 1:400; Abcam) and anti-human BNIP3 (catalog #44060, dilution 1:100; CST). After three washes (5 min each) in PBS, cells were incubated with secondary antibodies in blocking buffer for 30 min. The secondary antibodies used in immunofluorescence assays include donkey anti-mouse Alexa Fluor 647 (catalog #A-31571, dilution 1:400; Thermo Fisher Scientific) and donkey anti-rabbit Alexa Fluor 555 (catalog #A-31572, dilution 1:400; Thermo Fisher Scientific). Cells were subsequently washed three times in PBS before imaging. Images were acquired on a Nikon Ti2 microscope equipped with Yokogawa CSU-W1 spinning disk confocal and SoRa modules, a Hamamatsu Orca-Fusion sCMOS camera and a Nikon 60x objective (for Fig 3A and B) or Nikon 100x 1.45 NA (for Fig 4C and D) objective. All images were acquired using a 0.2-$\mu$m step size with the spinning disk module, and image adjustments were made with ImageJ/Fiji.

### DFO-induced iron chelation and washout

For immunoblotting experiments to visualize changes of BNIP3 and NIX protein levels upon iron chelation, 293T wild-type and *PPTC7* KO cells were treated with 100 $\mu$M DFO for 0–24 h as indicated in the results section. After the indicated incubation time, the cells were harvested by cell scraping in PBS before cell lysis and immunoblotting analysis. For qRT-PCR experiments to quantify changes in *BNIP3* and *BNIP3L* transcript level, 293T WT and *PPTC7* KO cells were treated with 100 $\mu$M DFO for 24 h, and the cells were harvested by cell scraping in PBS before RNA extracton and qRT-PCR analysis. For DFO washout experiments, after treating 293T WT and *PPTC7* KO cells with 100 $\mu$M DFO for 24 h, cells were washed with PBS and was switched to fresh DMEM media (for Fig 2B–F) or with DMEM containing 0.1% EtOH (as vehicle control) or 10 $\mu$M MG-132 (for Fig 2I) for 0–24 h as indicated in the results section. After the indicated incubation time, the cells were harvested by cell scraping in PBS before cell lysis and immunoblotting analysis. For flow cytometry experiments to determine the change of mitophagic flux with DFO dosage, MEF mt-Keima wild-type and *Pptc7* KO cells were treated with 0–100 $\mu$M DFO as indicated in the results section for 24 h before downstream processing for flow cytometry. To understand whether the DFO-induced mitophagy is BNIP3- and NIX-mediated, MEF mt-Keima wild-type, *Pptc7* KO, and *Pptc7/Bnip3/Bnip3l* TKO cells were treated with 100 $\mu$M DFO for 24 h before downstream processing for flow cytometry.

### MG-132 proteasomal inhibition time course

For MG-132 proteasomal inhibition time course experiment performed in Fig 2G and H, 293T WT and *PPTC7* KO cells were treated with 0.1% EtOH (as vehicle control) or 10 $\mu$M MG-132 for 0–6 h as indicated in the results section. After the indicated incubation time, the cells were harvested by cell scraping in PBS before cell lysis and immunoblotting analysis.

### Protease protection assay

HeLa FLP-IN TREx PPTC7-FLAG mt-Keima-positive cells were seeded at 1 × 10$^6$ in a 15-cm$^2$ plate. After 48 h, cells were treated with 10 $\mu$M doxycycline to induce PPTC7-FLAG expression for 40 h. Cells were then harvested by scraping in PBS and spun down at 600$g$ for 10 min at 4°C. To obtain a mitochondrial-enriched fraction, cells were resuspended in mitochondrial isolation buffer (0.1 M Tris-MOPS pH7.4, 0.1 M EGTA, 1 M sucrose in water) and homogenized at 1,000 rpm with a Potter-Elvehjem tissue homogenizer at 4°C. The resulting mixture was then centrifuged at 600$g$ for 10 min at 4°C, and the supernatant was then kept and recentrifuged at 7,000$g$ for 10 min at 4°C. The resulting pellet was the washed and resuspended with mitochondrial isolation buffer and quantified by using the Pierce 660 nm Protein Assay Kit (Thermo Fisher Scientific). After the mitochondrial-enriched fraction was obtained, 15 $\mu$g of crude mitochondria were used per condition. Proteinase K (10 $\mu$g/ml final concentration, NEB) and/or SDS (1% final, in 50 mM Tris–HCl, pH 8.0) were added as necessary per condition. The reaction mixtures were incubated on ice for 30 min, quenched by the addition of 5x sample buffer, and incubated at 95°C for 10 min. The samples were then loaded onto SDS–PAGE gels for downstream immunoblotting analyses.

### Analysis of PPTC7 localization relative to BNIP3

U2OS cells were transiently transfected with PPTC7-GFP plasmid, and then passaged into glass-bottom dishes. Cells were allowed to adhere for 12 h and subsequently treated with freshly prepared DFP (1 mM; Sigma-Aldrich) for 24 h before fixation. Where indicated, cells were then washed with fresh media, and fixed after an additional 4 h of incubation. Immunofluorescence staining and confocal microscopy were then performed on cells. To determine the co-enrichment of PPTC7 with BNIP3, enlarged foci enriched for BNIP3 signal relative to TOMM20 were manually counted in Fiji by examining single plane images throughout z-series of individual cells blinded to the corresponding PPTC7 image, followed by assessment of whether PPTC7 was co-enriched.

### Analysis of the effect of PPTC7 overexpression on BNIP3 immunofluorescence

For analysis of BNIP3 staining by immunofluorescence, HeLa cells were transiently transfected with cytosolic GFP or PPTC7-GFP

plasmids. Cells were allowed to adhere to glass-bottom dishes for 12 h and treated with freshly prepared $CoCl_2$ (500 $\mu M$; Sigma-Aldrich) 12 h before fixation. Cells were imaged with identical imaging conditions between experimental replicates. Cells that had GFP signal above an arbitrary threshold that was consistently maintained between experiments were blindly identified and manually categorized as having mitochondrial or diffuse non-mitochondrial BNIP3 signal in Fiji by examining maximum z-projections.

### Analysis of the effect of PPTC7 mutant expression on BNIP3 and NIX total protein levels

HeLa FLP-IN TREx cells expressing wild-type PPTC7 or various mutants (e.g., D78A) were generated as described above. To induce expression of these constructs, cells were treated with 10 $\mu M$ doxycycline for 24–32 h, after which BNIP3 and NIX expression was induced by treating cells with 200 $\mu M$ $CoCl_2$ for 16 h. Cells were collected, lysed in RIPA buffer, run via SDS–PAGE and western blotted for each protein as described above (see the "SDS–PAGE and immunoblotting" section).

### Proximity labeling

Wild-type 293T cells were transiently transfected with pcDNA3.1-MCS (empty vector), pcDNA3.1-V5-miniTurbo-NES, or pcDNA3.1-PPTC7-V5-miniTurbo before cells were treated with either vehicle only or 100 $\mu M$ DFO (supplemented in growth media) for another 24 h. For labeling, DMSO or 250 $\mu M$ biotin were added to cells, and cells were incubated at 37°C for 30 min. Labeling reactions were quenched by washing the cells three times with ice-cold PBS and incubating the cells at 4°C. Cells were then harvested in ice-cold PBS before lysed with RIPA, clarified, and quantified via bicinchoninic acid. The lysates were then incubated with 80 $\mu l$ streptavidin magnetic beads (New England Biolabs) at 4°C overnight. The beads were then washed twice with RIPA buffer, once with Wash Buffer 2 (500 mM sodium chloride, 0.1% wt/vol deoxycholate, 1% Triton X-100, 1 mM EDTA, 50 mM HEPES, pH 7.5), once with Wash Buffer 3 (250 mM sodium chloride, 0.5% Triton X-100, 0.5% wt/vol deoxycholate, 1 mM EDTA, 50 mM HEPES, pH 8.1), once with Wash Buffer 4 (150 mM sodium chloride, 50 mM HEPES, pH 7.4), and once with Wash buffer 5 (50 mM ammonium bicarbonate in MS-grade water). The bound species were eluted by adding 20 $\mu l$ 5x sample buffer, boiling at 95°C for 10 min, vortexing for 10 min, and boiling at 95°C for 10 min. The samples were then run on SDS–PAGE gels, and immunoblotting was performed to probe for BNIP3 and NIX.

### Data/statistical analysis and figure generation

Statistical analysis was performed using Microsoft Excel and/or Prism software (GraphPad, version 10). Fig S1A was generated using Seaborn library on Python. A selected list of HIF target genes were curated (Dengler et al, 2014) as mined from datasets in mouse *Pptc7* KO heart and liver proteomics (Niemi et al, 2019). Fig 2C was generated using AlphaFold2. Fig S3E was generated using AlphaFold

multimer mode; images were generated using PyMOL (Version 2.5.2). Fig 3A was generated using BioRender with an appropriate license. All figures were generated using Adobe Illustrator.

## Data Availability

No large datasets were generated in this article.

## Supplementary Information

## Acknowledgements

We thank Edrees Rashan, Keshav Kailash, Michael McKenna, and the Niemi Laboratory for careful reading and for helpful discussions on this work. We thank Julia Pagan (University of Queensland, Australia) for key resources and for important discussions related to this work. We thank the Flow Cytometry and Fluorescence-Activated Cell Sorting Core at Washington University School of Medicine for equipment and support for the flow cytometry experiments. This work was supported by R35GM151130 (to NM Niemi) and R35GM137894 (to JR Friedman). The UT Southwestern Quantitative Light Microscopy Facility, which is supported in part by NIH P30CA142543, provided access to the Nikon SoRa microscope (purchased with NIH 1S10OD028630-01). L Wei was supported by the MilliporeSigma Predoctoral Fellowship in Honor of Dr. Gerty T. Cori at Washington University. JD Svoboda was supported through the SURGE Summer Undergraduate Research Program put on within the Department of Biochemistry and Molecular Biophysics at Washington University School of Medicine in St. Louis.

### Author Contributions

L Wei: conceptualization, data curation, formal analysis, investigation, methodology, project administration, and writing—original draft, review, and editing.
MO Gok: data curation, formal analysis, and writing—review and editing.
JD Svoboda: data curation, formal analysis, and writing—review and editing.
K-L Kozul: resources, data curation, formal analysis, and writing—review and editing.
M Forny: resources, data curation, formal analysis, funding acquisition, project administration, and writing—original draft, review, and editing.
JR Friedman: conceptualization, data curation, formal analysis, supervision, funding acquisition, investigation, project administration, and Writing—original draft, review, and editing.
NM Niemi: conceptualization, data curation, formal analysis, supervision, funding acquisition, investigation, project administration, and writing—original draft, review, and editing.

### Conflict of Interest Statement

The authors declare that they have no conflict of interest.

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
