## [Reviewer comments · Life Science Alliance]

Life Science Alliance

Dual localized PPTC7 limits mitophagy through proximal and dynamic interactions with BNIP3 and NIX

Lianjie Wei, Mehmet Gok, Jordyn Svoboda, Keri-Lyn Kozul, Merima Forný, Jonathan Friedman, and Natalie Niemi
DOI: <https://doi.org/10.26508/lsa.202402765>

Corresponding author(s): Natalie Niemi, Washington University in St. Louis School of Medicine

Review Timeline:	Submission Date:	2024-04-11
	Editorial Decision:	2024-04-22
	Revision Received:	2024-06-07
	Editorial Decision:	2024-06-10
	Revision Received:	2024-06-10
	Accepted:	2024-06-11

Transaction Report:

Please note that the manuscript was previously reviewed at another journal and the reports were taken into account in the decision-making process at *Life Science Alliance*.

Reviews

Referee #1

Report for Author:

In previous work, Niemi et al (Nat. Comm. 2023) showed that loss of the PPTC7 phosphatase promoted mitophagy in a manner dependent on BNIP3 and NIX and showed PPTC7 to interact directly with BNIP3 and NIX, presumably to inhibit their activity.

Recently Sun et al (Mol. Cell 2024) showed that a full-length form of PPTC7 promoted the FBXL4-dependent degradation of BNIP3 and NIX independent of its phosphatase activity by forming a complex with BNIP3 or NIX and FBXL4 at the outer mitochondrial membrane and that loss of NIX rescued the perinatal lethality of PPTC7 knockout.

Here the authors show that the effect of PPTC7 on levels of BNIP3 and NIX is at the post-transcriptional level and depends on proteasomal degradation. The authors also argue that the active site of the PPTC7 enzyme is required to turnover BNIP3 and NIX since the D78A mutant fails to suppress their expression in contrast to wild-type PPTC7. However as the authors themselves point out this could be explained by the failure of the D78A mutant to bind to BNIP3/NIX, as opposed to a reliance on its phosphatase activity. They then show that removing its MTS and artificially targeting PPTC7 to the OMM permits it to regulate BNIP3 and NIX levels, and via proximity ligation shows that PPTC7 co-localizes with BNIP3 and NIX, arguing similar to Sun et al, that PPTC7 regulates BNIP3 and NIX at the OMM and not in the matrix where PPTC7 is otherwise located.

Major Issues.

1. The major issue is the lack of novelty of this work compared to previous work from the senior author (Niemi et al, 2023) and most particularly compared to the work reported in Molecular Cell paper by Sun et al (2024). The authors really do not show anything new here that is not already reported elsewhere. The Sun et al work is considerably more extensive and complete. Even the data showing the effect is post-transcriptional is not novel since Nguyen-Dien et al (2023) showed effect was HIF1-independent.

2. Critically, the reader is left not knowing whether the phosphatase activity of PPTC7 is required for the ability to promote BNIP3 and NIX turnover since the D78A mutant may affect binding to BNIP3 and NIX in addition to blunting its phosphatase activity. To address this, Sun et al mutated key residues in BNIP3 and NIX and showed that this did not affect the ability of PPTC7 to promote BNIP3 and NIX turnover arguing that the phosphatase activity of PPTC7 is NOT required. Absent further analysis here to address this question further, the current work is inconclusive.

Minor Issues.

1. Park et al, 2013 and Poole et al, 2021 cited in the text (line 189) are missing from the references.

Referee #2

Report for Author:

This study by Wei and colleagues investigates the interconnections between the mitochondrial phosphatase PPTC7 and its ability to regulate mitophagy. Specifically, the authors show that PPTC7 loss leads to stabilization of NIX and BNIP3 enhancing mitophagy, data presented show that this is likely through an ability of PPTC7 to promote proteasome dependent degradation of NIX and BNIP3, the authors proceed to identify two different forms of PPTC7, one within the mitochondria and one at the OMM, showing that cytosolic PPTC7 (or mito OMM targeted) is sufficient to promote BNIP3, NIX degradation. Their data supports a role for PPTC7 phosphatase activity and a direct interaction with BNIP3 and NIX in promoting their degradation. Overall the authors present an interesting and timely study that nicely complements the recently published work in this area (cited within). However, I think key outstanding questions remain that should be addressed. These are detailed below.

- is this pathway (stability regulation by PPTC7) dependent on FBXL4 ? i.e. does deletion of FBXL4 stabilize NIX/BNIP3 further (suggesting independent pathways) or have no effect (suggesting the same pathway) in PPTC7 deleted cells ?

- is the long-form of PPTC7 indeed localized to the mitochondrial OMM (as the authors suggest?), this could be addressed by limited proteolysis expts.

- is the import of PPTC7 affected under hypoxic conditions, e.g. could the OMM form be induced to counteract NIX, BNIP mediated mitophagy ?

- the contradiction between the impact of loss of phosphatase function here on NIX, BNIP stabilization and with what has recently been published is quite striking, potentially the phosphatase dead mutant of PPTC7 can no longer interact with NIX, BNIP3, explaining the contradiction - this should be examined.

- the stabilization expt in 2I is a bit unclear to me, it seems like the MG132 significantly stabilizes in WT cells even after washout, which at that point should be removing the transcriptional upregulation of these proteins (i.e. the MG132 should only stabilize what is already present) this doesn't appear the case, the authors should clarify

April 22, 2024

Re: Life Science Alliance manuscript #LSA-2024-02765-T

Dr. Natalie M. Niemi
Washington University in St. Louis
Biochemistry & Molecular Biophysics
Unknown
St. Louis, MO 63110

Dear Dr. Niemi,

Thank you for submitting your manuscript entitled "PPTC7 limits mitophagy through proximal and dynamic interactions with BNIP3 and NIX" to Life Science Alliance. Please address the Reviewers' comments via added Discussion points and clarification. If any data already exists to address any of the points, that data should be included in the revision.

Thank you for this interesting contribution to Life Science Alliance. We are looking forward to receiving your revised manuscript.

Sincerely,

B. MANUSCRIPT ORGANIZATION AND FORMATTING:

Manuscript LSA-2024-02765-T

PPTC7 limits mitophagy through proximal and dynamic interactions with BNIP3 and NIX

Wei et al.

Response to reviewers

We thank both reviewers for their comments and lines of questioning, which we feel have elevated our manuscript and its findings. We are providing a point-by-point response to each of the reviewers' comments, shown in blue below.

Reviewer #1

The major issue is the lack of novelty of this work compared to previous work from the senior author (Niemi et al, 2023) and most particularly compared to the work reported in Molecular Cell paper by Sun et al (2024). The authors really do not show anything new here that is not already reported elsewhere. The Sun et al work is considerably more extensive and complete. Even the data showing the effect is post-transcriptional is not novel since Nguyen-Dien et al (2023) showed effect was HIF1-independent.

We respectfully disagree with this comment. Our previous work (Niemi et al., *Nat Commun* 2023) established a link between PPTC7 and the mitophagy receptors BNIP3 and NIX. However, it remained unclear how PPTC7 could influence BNIP3 and NIX, as these two proteins were thought to reside in separate compartments. This work directly addresses this question by uncovering important mechanistic insights regarding the regulation of PPTC7 on BNIP3/NIX.

First, we demonstrate that PPTC7 influences BNIP3/NIX independent of HIF-1. While reviewer #1 claims that Nguyen-Dien et al. showed this in their 2023 EMBO paper, critically, this paper characterizes FBXL4 *only* and has no mention of PPTC7. Given the extensive literature regarding BNIP3/NIX transcriptional regulation, our data demonstrating that *PPTC7* KO cells increase BNIP3/NIX levels independent of HIF-1 were key to establishing insights into its mechanism. Furthermore, our data demonstrate that pseudohypoxic activation of HIF-1 further increases BNIP3/NIX protein expression in *PPTC7* KO cells, establishing that BNIP3/NIX are not maximally expressed upon loss of PPTC7. These data further indicate that *PPTC7* KO cells and tissues are likely susceptible to hypoxic environments due to these additive effects. As such, these are valuable insights for understand the contexts in which PPTC7 may contribute to pathophysiology in the future.

Second, we establish a DFO wash-out system for interrogating the dynamics and turnover rates of endogenous BNIP3 and NIX in cells. Importantly, our gel resolving system allows us to separate and quantify both monomeric and dimeric populations of BNIP3 and NIX, and our data suggest distinct half-lives for each of these species. Furthermore, our data link ubiquitination-mediated turnover pathways to BNIP3/NIX in *PPTC7* KO cells – a finding that is consistent with the recent Sun et al. *Mol Cell* 2023 paper, as well as a recent preprint from the Pagan lab (Nguyen-Dien et al., bioRxiv, 2024).

Third, we demonstrate that PPTC7 interacts with BNIP3 and NIX in cells using proximity labeling assays. While the Sun et al., *Mol Cell* 2023 and Nguyen-Dien et al., bioRxiv, 2024 studies have used recombinant proteins and/or immunoprecipitations to demonstrate interactions between PPTC7 and BNIP3/NIX, these methods cannot demonstrate that these interactions occur within the native cellular environment. Given the potentially disparate compartmentalization of these proteins, these insights were critical for understanding PPTC7-BNIP3/NIX. Thus our proximity labeling studies nicely complement these two studies, which collectively suggest that PPTC7 directly binds to BNIP3/NIX in native cellular conditions. We complement these assays by demonstrating that GFP-PPTC7 co-localizes to BNIP3-positive punctae that likely represent nascent mitophagosomes. Finally, we exploit the DFO-washout system to demonstrate that the PPTC7-BNIP3/NIX interactions and co-localization in cells are enriched acutely after triggering BNIP3/NIX turnover. These data

suggest that PPTC7 is dynamically recruited to facilitate BNIP3/NIX turnover, with such insights critical for understanding how PPTC7 regulates mitophagy.

Critically, the reader is left not knowing whether the phosphatase activity of PPTC7 is required for the ability to promote BNIP3 and NIX turnover since the D78A mutant may affect binding to BNIP3 and NIX in addition to blunting its phosphatase activity. To address this, Sun et al mutated key residues in BNIP3 and NIX and showed that this did not affect the ability of PPTC7 to promote BNIP3 and NIX turnover arguing that the phosphatase activity of PPTC7 is NOT required. Absent further analysis here to address this question further, the current work is inconclusive.

We agree that this is an important point, and one which we have thought about extensively. Unfortunately, the emerging evidence from our lab, as well as the Jiang and Pagan labs, suggest a complex model of PPTC7-mediated regulation of BNIP3/NIX, which involves multiple binding partners and dynamic recruitment. Given these recent insights, we believe that pinpointing the exact mechanism by which the D78A mutant disallows BNIP3/NIX turnover is beyond the scope of this manuscript. We consider the complexities of the model that has led us to this conclusion below.

Our previous paper (Niemi et al, *Nat Commun* 2023), as well as data in this paper (Supplemental Figure 3C) demonstrate that recombinant PPTC7 can directly dephosphorylate BNIP3 and NIX, whereas the D78A mutant fails to do so. Importantly, BNIP3/NIX are hyperphosphorylated in *PPTC7* KO cells, and previous work (He et al., *Cell Death Disease* 2022) demonstrates that phosphorylation at least some of these residues can increase the stability of BNIP3/NIX in cells. The most straightforward model stemming from these data is that PPTC7 directly dephosphorylates BNIP3/NIX to promote their degradation. Consistent with this model, our catalytically inactive mutant, D78A, cannot suppress BNIP3/NIX accumulation as shown in our manuscript (Figure 3F).

The studies from the Jiang and Pagan labs, however, suggest a more complicated model. The Sun et al. paper did not mutate key residues in BNIP3/NIX, but rather used CdCl₂ to globally inhibit PP2C phosphatases to suggest that the phosphatase activity of PPTC7 is not required for BNIP3/NIX turnover. However, it is not clear that CdCl₂ fully inhibits PPTC7 in these experiments, and the dosing of PPTC7 within cells may be complicated by factors such as efficiency of CdCl₂ cellular uptake. A second piece of evidence that PPTC7 may not require phosphatase activity comes from the Nguyen-Dien et al., bioRxiv, 2024 study. In this preprint, the authors generated point mutants of PPTC7 to disrupt phosphatase activity without affecting binding to BNIP3/NIX based on their structural modeling. One of these mutants, D290N, has substantially reduced phosphatase activity but still allows turnover of BNIP3/NIX. However, it is unclear whether the phosphatase activity in the D290N is fully ablated, or is rather significantly blunted, and whether such distinctions in activity would matter. It is possible that remnant enzymatic activity, albeit inefficient, could still allow PPTC7 to facilitate BNIP3/NIX turnover. However, given these data, particularly in light of the rigorous structural characterization performed in this paper, as well as the data from the Jiang lab, it is possible if not likely that PPTC7 phosphatase activity is not required for the turnover of BNIP3/NIX.

The mechanism by which D78A perturbs PPTC7 function is likely complex, based on studies from all three groups. First, the Nguyen-Dien study demonstrated that PPTC7 D78A does not efficiently immunoprecipitate BNIP3/NIX. These data led us to consider that our mutant may be non-functional not because of a loss of phosphatase activity, but because it may not bind BNIP3/NIX to the same extent that wild-type PPTC7 does, which we have shown through additional data in our manuscript (Supplementary Figure 3F). Notably, however, the Sun et al. study uses a mutant similar to the mutant we used (i.e., Sun et al. use a PPTC7 2A mutant, which is mutated at both D78A/G79A, our construct is a single D78A mutant). Sun et al. demonstrate that the PPTC7 2A D78A/G79A mutant maintains direct binding to BNIP3/NIX using recombinant proteins, but disrupts binding to a Cullin1-Rbx1 complex – critical protein within the SCF complex that facilitate FBXL4-mediated ubiquitination. While these data seem conflicting, it is important to consider the differences in experimental systems used across these papers in generating these conclusions. The Sun et al. paper uses recombinant proteins isolated from *E. coli* to demonstrate that the BNIP3/NIX-PPTC7 D78A/G79A mutant interaction is as efficient as the interaction with wild-type PPTC7. However, both our study, as well as the Nguyen-Dien study,

use immunoprecipitations from mammalian cells to probe these interactions. Collectively, these data suggest that PPTC7 D78A can bind to BNIP3/NIX in vitro, but cannot engage these receptors in cells. Interestingly, the Sun et al. study showed that the PPTC7 D78A/G79A mutant disrupts binding to the Cullin-Rbx1 complex in vitro, which brings forth a number of interesting models. The first is that PPTC7 facilitates BNIP3/NIX degradation through stepwise interactions with multiple partners, and that disruption of one of these interactions (e.g., the Cullin-Rbx1 complex) disallows functional complex assembly in vitro. The second possibility is that the PPTC7 D78A/G79A mutant has disrupted binding due to a required dephosphorylation event within the Cullin-Rbx1 SCF subcomplex. Given that these experiments were performed using recombinant proteins, we would typically dismiss this model, as post-translational modifications are usually not present on recombinant proteins. However, it is notable that the Sun et al. study purified recombinant Cullin1-Rbx1 protein complex from a mammalian source (Exp1293F cells), suggesting that relevant post-translational modifications may be present on these proteins to influence their assembly and function in vitro.

In light of these observations, it will be critical to examine multiple phosphatase-dead PPTC7 mutants, including the ones published across these studies, for their effects on PPTC7 phosphatase activity and binding to BNIP3/NIX as well as other SCF components. These studies will be critical to understand how such disparate phenotypes are seen between the D78A and D290N PPTC7 mutants. Unfortunately, given our lack of detailed knowledge about how these multiprotein subunit complexes interact and how these interactions influence BNIP3/NIX turnover, we propose that these experiments are beyond the scope of this manuscript. We are, however, very interested in this question, and plan to perform these detailed studies in future work.

Reviewer #2

This study by Wei and colleagues investigates the interconnections between the mitochondrial phosphatase PPTC7 and its ability to regulate mitophagy. Specifically, the authors show that PPTC7 loss leads to stabilization of NIX and BNIP3 enhancing mitophagy, data presented show that this is likely through an ability of PPTC7 to promote proteasome dependent degradation of NIX and BNIP3, the authors proceed to identify two different forms of PPTC7, one within the mitochondria and one at the OMM, showing that cytosolic PPTC7 (or mito OMM targeted) is sufficient to promote BNIP3, NIX degradation. Their data supports a role for PPTC7 phosphatase activity and a direct interaction with BNIP3 and NIX in promoting their degradation. Overall the authors present an interesting and timely study that nicely complements the recently published work in this area (cited within). However, I think key outstanding questions remain that should be addressed. These are detailed below.

We thank the reviewer for their positive comments, as well as their insightful questions, which we address below.

- is this pathway (stability regulation by PPTC7) dependent on FBXL4 ? i.e. does deletion of FBXL4 stabilize NIX/BNIP further (suggesting independent pathways) or have no effect (suggesting the same pathway) in PPTC7 deleted cells ?

The dependence of BNIP3/NIX has been shown to be dependent on FBXL4 in both the recent Sun et al., Mol Cell 2023 and Nguyen-Dien et al., bioRxiv, 2024 studies. We address these studies and acknowledge this work in our discussion.

- is the long-form of PPTC7 indeed localized to the mitochondrial OMM (as the authors suggest?), this could be addressed by limited proteolysis expts.

This is an excellent suggestion. We have performed this experiment, which can be found in Figure 3E in our revised manuscript. As the reviewer suggests, we find that full length PPTC7 is susceptible to protease digestion, whereas the processed form of PPTC7 is protected. These experiments are consistent with our proposed model, as well as data from the Sun et al., Mol Cell 2023 and Nguyen-Dien et al., bioRxiv, 2024 studies.

- is the import of PPTC7 affected under hypoxic conditions, e.g. could the OMM form be induced to counteract NIX, BNIP mediated mitophagy ?

This is an interesting question. Indeed, our data suggest that the dual localization of PPTC7 is dynamic, as CoCl_2 treatment of wild-type PPTC7 causes an increase in full length, but not processed, PPTC7 (see lanes 4 of Figures 3F, G, and H). However, whether this is mediated at the level of import is not clear. We have addressed the dynamic behavior between PPTC7 isoforms, and what might be influencing it, within the discussion of our revised manuscript.

- the contradiction between the impact of loss of phosphatase function here on NIX, BNIP stabilization and with what has recently been published is quite striking, potentially the phosphatase dead mutant of PPTC7 can no longer interact with NIX, BNIP3, explaining the contradiction - this should be examined.

We agree, but, as stated above, the model of how this mutant interferes with BNIP3/NIX degradation is likely more complex than originally anticipated. We discuss these caveats, as well as what is currently known given our study as well as the Jiang and Pagan studies, in our revised manuscript.

- the stabilization expt in 2I is a bit unclear to me, it seems like the MG132 significantly stabilizes in WT cells even after washout, which at that point should be removing the transcriptional upregulation of these proteins (i.e. the MG132 should only stabilize what is already present) this doesn't appear the case, the authors should clarify

We believe this is the case because there is basal transcription of BNIP3 and NIX, and it is possible that the decrease in HIF1-mediated transcription of BNIP3 and NIX upon washout is gradual. As the MG-132 treatment can also stabilize BNIP3 and NIX synthesized during the washout, this would lead to a significant stabilization of BNIP3 and NIX in wild-type cells after the washout.

June 10, 2024

RE: Life Science Alliance Manuscript #LSA-2024-02765-TR

Dr. Natalie M. Niemi
Washington University in St. Louis School of Medicine
Biochemistry & Molecular Biophysics
Unknown
St. Louis, MO 63110

Dear Dr. Niemi,

Thank you for submitting your revised manuscript entitled "Dual localized PPTC7 limits mitophagy through proximal and dynamic interactions with BNIP3 and NIX". We would be happy to publish your paper in Life Science Alliance pending final revisions necessary to meet our formatting guidelines.

- please be sure that the authorship listing and order is correct
- please add a Running Title to our system
- please add an Abstract and Summary Blurb/Alternate Abstract to our system
- please add the Twitter handle of your host institute/organization as well as your own or/and one of the authors in our system
- please consult our manuscript preparation guidelines <https://www.life-science-alliance.org/manuscript-prep> and make sure your manuscript sections are in the correct order
- please add an Author Contributions section to your main manuscript text
- please add your main and supplementary figure legends to the main manuscript text after the references section
- please add callouts for Figure S3A-F to your main manuscript text

A. FINAL FILES:

B. MANUSCRIPT ORGANIZATION AND FORMATTING:

Sincerely,

June 11, 2024

RE: Life Science Alliance Manuscript #LSA-2024-02765-TRR

Dr. Natalie M. Niemi
Washington University in St. Louis School of Medicine
Biochemistry & Molecular Biophysics
660 Euclid Ave, Box 8231
St. Louis, MO 63110

Dear Dr. Niemi,

Thank you for submitting your Research Article entitled "Dual localized PPTC7 limits mitophagy through proximal and dynamic interactions with BNIP3 and NIX". It is a pleasure to let you know that your manuscript is now accepted for publication in Life Science Alliance. Congratulations on this interesting work.

DISTRIBUTION OF MATERIALS:

Again, congratulations on a very nice paper. I hope you found the review process to be constructive and are pleased with how the manuscript was handled editorially. We look forward to future exciting submissions from your lab.

Sincerely,
